# NEURAL SPATIO-TEMPORAL REASONING WITH OBJECT-CENTRIC SELF-SUPERVISED LEARNING

## ABSTRACT

Transformer-based language models have proved capable of rudimentary symbolic reasoning, underlining the effectiveness of applying self-attention computations to sets of discrete entities. In this work, we apply this lesson to videos of physical interaction between objects. We show that self-attention-based models operating on discrete, learned, object-centric representations perform well on spatio-temporal reasoning tasks which were expressly designed to trouble traditional neural network models and to require higher-level cognitive processes such as causal reasoning and understanding of intuitive physics and narrative structure. We achieve state of the art results on two datasets, CLEVRER and CATER, significantly outperforming leading hybrid neuro-symbolic models. Moreover, we find that techniques from language modelling, such as BERT-style semi-supervised predictive losses, allow our model to surpass neuro-symbolic approaches while using 40% less labelled data. Our results corroborate the idea that neural networks can reason about the causal, dynamic structure of visual data and attain understanding of intuitive physics, which counters the popular claim that they are only effective at perceptual pattern-recognition and not reasoning per se.

## 1 INTRODUCTION

Artificial intelligence research has long been divided into rule-based approaches and statistical models. Neural networks, a classic example of the statistical approach, certainly have limitations despite their massive popularity and success. For example, experiments with two recently released video question-answering datasets, CLEVRER (Yi et al., 2020) and CATER (Girdhar & Ramanan, 2020), demonstrate that neural networks fail to adequately reason about spatio-temporal and compositional structure in visual scenes. While the networks perform adequately when asked to *describe* their inputs, they tend to fail when asked to *predict*, *explain*, or consider *counterfactual* possibilities.

By contrast, a neuro-symbolic model called NS-DR (Yi et al., 2020) appears to be much better suited to predicting, explaining, and considering counterfactual possibilities with this data. The model leverages independent neural networks to detect objects, infer dynamics, and syntactically parse the question. A hand-coded symbolic executor interprets the questions grounded on the outputs of the networks. The fact that hybrid models employing both distributed (neural) representations and symbolic logic can sometimes perform better has led some to consider neuro-symbolic hybrids to be a more promising model class compared to end-to-end neural networks (Andreas et al., 2016; Yi et al., 2018; Marcus, 2020).

There is evidence from other domains, however, that neural networks can indeed adequately model higher-level cognitive processes. For example, in some symbolic domains (such as language), neural networks outperform hybrid neuro-symbolic approaches when tasked to classify or predict (Devlin et al., 2018). Neural models have also had some success in mathematics, a domain that, intuitively, would seem to require the execution of formal rules and manipulation of symbols (Lample & Charton, 2020). Somewhat surprisingly, large-scale neural language models such as GPT-3 (Brown et al., 2020) can acquire a propensity for arithmetic reasoning and analogy-making without being trained explicitly for such tasks, suggesting that current neural network limitations are ameliorated when scaling to more data and using larger, more efficient architectures (Brown et al., 2020; Mitchell, 2020). A key motivation of our work, therefore, is to reconcile existing neural network limitations in video domains with their (perhaps surprising) successes in symbolic domains.

One common element of these latter results is the repeated application of self-attention processes (Vaswani et al., 2017) to sequences of discrete 'entities'. Here, we apply this insight to videos of physical interactions between sets of objects, where the input data to models are continuously-valued pixel arrays at multiple timesteps (together with symbolic questions in certain cases).

A key design decision is the appropriate level of granularity for the discrete units underlying the self-attention computation. What is the visual analogue to a word in language, or a symbol in mathematics? We hypothesize that the discrete entities acted upon by self-attention should correspond to semantic entities relevant to the task. For tasks based on visual data derived from physical interactions, these entities are often times *objects* (van Steenkiste et al., 2019; Battaglia et al., 2018). To extract representations of these entities, we use MONet, an unsupervised object segmentation model (Burgess et al., 2019), but we leave open the possibility that other object-estimation algorithms might work better. We propose that a sufficiently expressive self-attention model acting on entities corresponding to physical objects will exhibit, on video datasets, a similar level of higher-level cognition and 'reasoning' seen when these models are applied to language or mathematics.

Altogether, our results demonstrate that self-attention-based neural nets can outperform hybrid neuro-symbolic models on visual tasks that require high-level cognitive processes, such as causal reasoning and physical understanding. We show that choosing the right level of discretization is critical for successfully learning these higher-order capabilities: pixels and local features are too fine, and entire scenes are too coarse. Moreover, we identify the value of self-supervised tasks, especially in low data regimes. These tasks ask the model to infer future arrangements of objects given the past, or to infer what must have happened for objects to look as they do in the present. We verify these conclusions in two video datasets, one in which the input is exclusively visual (CATER) and one that requires the combination of language (questions) and vision (CLEVRER).

## 2 METHODS

Our principal motivation is the converging evidence for the value of self-attention mechanisms operating on a finite sequences of discrete entities. Written language is inherently discrete and hence is well-suited to self-attention-based approaches. In other domains, such as raw audio or vision, it is less clear how to leverage self-attention. We hypothesize that the application of self-attention-based models to visual tasks could benefit from an approximate 'discretization' process analogous to the segmentation of speech into words or morphemes, and that determining the appropriate level of discretization is an important choice that can significantly affect model performance.

At the finest level, data could simply be discretized into pixels (as is already the case for most machine-processed visual data). But since pixels are too-fine grained, some work considers the downsampled "hyper-pixel" outputs of a convolutional network to comprise the set of discrete units (e.g. Zambaldi et al. (2019); Lu et al. (2019)). In the case of videos, an even courser discretization scheme is often used: representations of frames or subclips (Sun et al., 2019b).

The neuroscience literature, however, suggests that biological visual systems infer and exploit the existence of *objects*, rather than use spatial or temporal blocks with artificial boundaries (Roelfsema et al., 1998; Spelke, 2000; Chen, 2012). Because objects are the atomic units that tasks we consider here focus on, it makes sense to discretize on the level of objects. Numerous object segmentation algorithms have been proposed (Ren et al., 2015; He et al., 2017; Greff et al., 2019). We chose to use MONet, an unsupervised object segmentation algorithm that produces object representations with disentangled features (Burgess et al., 2019). Because MONet is unsupervised, we can train it directly in our domain of interest without the need for object segmentation labels.

To segment each frame into object representations, **MONet** first uses a recurrent attention network to obtain a set of $N_o$ "object attention masks" ($N_o$ is a fixed parameter). Each attention mask represents the probability that any given pixel belongs to that mask's object. The pixels assigned to the mask are encoded into latent variables with means $\mu_{ti} \in \mathbb{R}^d$, where $i$ indexes the object and $t$ the frame. These means are used as the object representations in our model. More details are provided in Appendix A.1.

The **self-attention component** is a transformer model (Vaswani et al., 2017) over the sequence $\mu_{ti}$. In addition to this sequence of vectors, we include a trainable vector $CLS \in \mathbb{R}^d$ that is used to generate classification results; this plays a similar role to the CLS token in BERT (Devlin et al., 2018). Finally, for our CLEVRER experiments, where the inputs include a question and potentially several

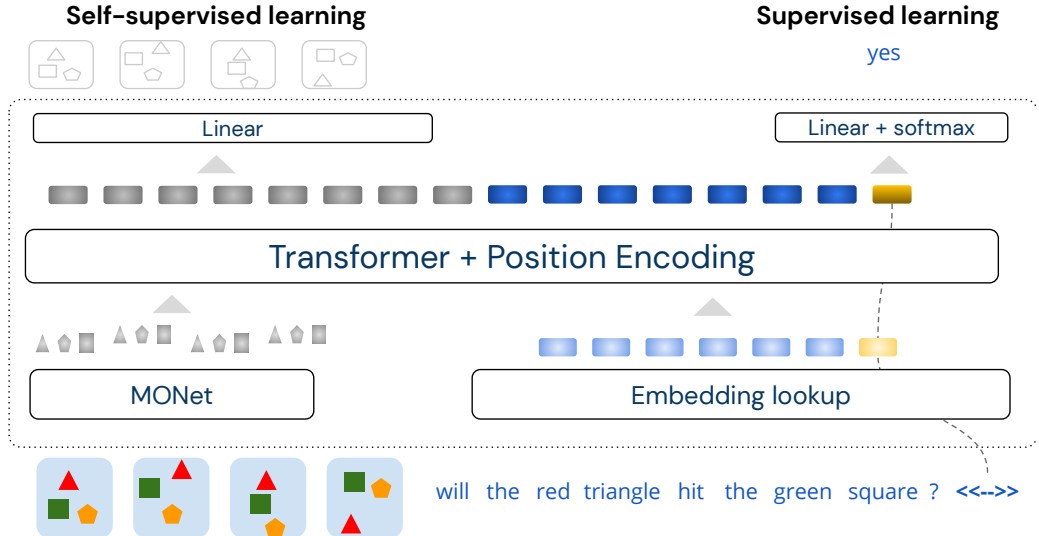

Figure 1: A schematic of the model architecture. See the main text for details.

choices, we embed each question word $\mathbf{w}_i$ (and choice for multiple choice questions) in $\mathbb{R}^d$ and include these in the sequence of inputs to the transformer. We also append a two-dimensional one-hot vector to $\mu_{ti}$ and $\mathbf{w}_i$ to indicate whether the input is a word or an object.

We pass the sequence consisting of the object latent means $\mu_{ti}$, the classification token, and the word embeddings (for CLEVRER) through a transformer with $N_T$ layers. We add a relative positional encoding at each layer of the transformer to give the model knowledge of the word and frame order (Dai et al., 2019). The transformed value of the classification token $CLS$ is passed through an MLP (with one hidden layer of size $N_H$) to generate the final answer.

This general approach is adapted to each of our datasets according to the format of expected answers. The final layer, which operates on the transformed value of $CLS$, is a softmax over answers for CLEVRER descriptive questions, a logit for each choice for CLEVRER multiple-choice questions, or a softmax over the grid-index of the final location of the snitch for CATER. A schema of our architecture is shown in Figure 1.

Note that in this model, an object in one frame can attend to every object in every frame. We also consider an alternative model with *hierarchical attention*, which consists of two stages of self-attention with two different transformers. The first transformer acts independently on the objects within each frame along with the word embeddings. The outputs of the first transformer for each frame are concatenated into a single feature vector, one for each frame. The second transformer acts on these feature vectors, treating each frame as an atomic entity. We study the importance of global attention (objects as the atomic entities) vs hierarchical attention (objects, and subsequently frames as the atomic entities). The comparison is shown in Table 1.

## 2.1 SELF-SUPERVISED LEARNING

Self-supervised learning—unsupervised learning where the data provides its own supervision—has a long history in language processing, where it allows models to acquire useful representations of words, phrases and sentences (Mikolov et al., 2013; Kiros et al., 2015; Devlin et al., 2018). Such techniques are also effective in visual domains, for image classification (Chen et al., 2020), video analysis (Han et al., 2019), and RL agents (Gregor et al., 2019). In these cases, it is proposed that the learned representations are more informationally dense, allowing models to satisfy their ultimate objectives more effectively. MONet, the technique we use to estimate object-centric visual representations, can also be considered a self-supervised learning method.

We explored whether self-supervised learning could improve the performance of our model beyond the benefits conveyed by object-level representation, i.e. in ways that support the model's interpretation

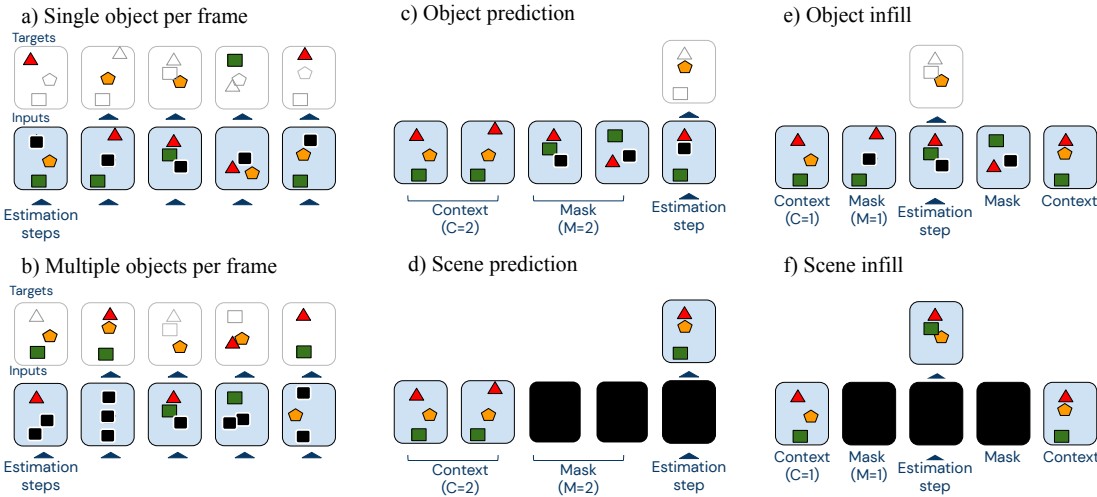

Figure 2: Different masking schemes for self-supervised learning applied to our model.

of the dynamics of the scenes rather than just via improved perception of static observations. Our approach is inspired by the loss used in BERT (Devlin et al., 2018), where a bidirectional transformer model is trained to predict certain words that are masked from the input. In our case, we mask *object representations*, and train the model to infer the content of the masked object representations using its knowledge from all unmasked objects.

More formally, we set

$$\text{transformer input} = \left\langle CLS; \, m_{ti}\mu_{ti}|_{t,i} \, ; \, \mathbf{w}_i|_i \right\rangle,$$

where $m_{ti} \in \{0, 1\}$ is a masking indicator. We write the output of the transformer as

$$\text{transformer output} = \left\langle CLS'; \, \mu'_{ti}|_{t,i} \, ; \, \mathbf{w}'_i|_i \right\rangle.$$

We expect the transformer to understand the underlying dynamics of the video, so that the masked out slot $\mu_{ti}$ could be predicted from $\mu'_{ti}$. We add an auxiliary loss to guide the transformer in learning effective representations capable of this type of dynamics prediction:

$$\text{auxiliary loss} = \sum_{t,i} \tau_{ti} l \left( f(\mu'_{ti}), \mu \right),$$

where $f$ is a learned linear mapping to $\mathbb{R}^d$, $l$ a loss function, and $\tau_{ti} \in \{0, 1\}$ are one-hot indicator variables identifying the prediction targets. We propagate gradients only to the parameters of $f$ and the transformer. This auxiliary loss is added to the main classification loss with weighting $\lambda$, and both losses are minimized simultaneously by the optimizer. In particular, we do not pretrain the model with only the auxiliary loss.

We tested two different loss functions, an L2 loss and a contrastive loss (formulas given in Appendix A.2), and six different masking schemes, as illustrated in Figure 2. This exploration was motivated by the observation that video inputs at adjacent timesteps are highly correlated in a way that adjacent words are not. We thus hypothesized that BERT-style prediction of adjacent words might not be optimal. A different masking strategy, in which prediction targets are separated from the context by more than a single timestep, may stimulate capacity in the network to acquire the environment knowledge that permits context-based unrolls and better long-horizon predictions.

First, we set $m_{ti} = 1$ (uniformly) at random across $t$ and $i$ and $\tau_{ti} = 1 - m_{ti}$, generated by fixing the expected proportion of the $m_{ti}$ set to 1 (schema $b$ in Figure 2). While simple, this has the downside of masking out multiple objects per frame, which is potentially problematic since MONet does not assign objects to slot indices in a well defined way. MONet usually switches object-to-slot

| Model | Descriptive | Explanatory | Predictive | Counterfactual |
|---|---|---|---|---|
| MAC (V+) | 86.4 | 22.3 | 42.9 | 25.1 |
| NS-DR | 88.1 | 79.6 | 68.7 | 42.2 |
| DCL | 90.7 | 82.8 | 82.0 | 46.5 |
| Ours | $\mathbf{94.0} \pm 0.4$ | $\mathbf{96.0} \pm 0.6$ | $\mathbf{87.5} \pm 3.0$ | $\mathbf{75.6} \pm 3.8$ |
| Ours − self-attention + MLP | 45.4 | 16.0 | 27.7 | 9.9 |
| Ours − object-repr. + ResNet | 62.3 | 54.1 | 46.2 | 23.7 |
| Ours − hierarchical + global attn. | 80.6 | 87.4 | 73.5 | 55.1 |
| Ours − self-supervised loss | 91.0 | 92.8 | 82.8 | 68.7 |

Table 1: Performance (per question accuracy) on CLEVRER of our model compared to results from literature and to ablations: 1) MLP instead of self-attention; 2) ResNet superpixels instead of MONet objects; 3) hierarchical frame-level and intra-frame attention instead of global cross-frame object attention; 4) no auxiliary loss.

assignments multiple times in a single video, and these switches occur unpredictably. If multiple slots are masked out, the transformer cannot determine with certainty which missing object to assign to each slot, and so the auxiliary loss could penalize the model even if it predicted all the objects correctly. To avoid this problem, we also try constraining the mask such that exactly one $m_{ti} = 0$ for each $t$ (schema $a$); this ensures only one slot per frame is masked out, eliminating the ambiguity.

To pose harder prediction challenges, we add a buffer between the context (where $m_{ti} = 1$) and the infilling targets (where $\tau_{ti} = 1$). For $t$ in this buffer zone, both $m_{ti} = 0$ and $\tau_{ti} = 0$ (schemas $c$–$f$). In the presence of this buffer, we compared prediction (where the context is strictly before the targets; schema $c$, $d$) versus infilling (where the context surrounds the targets; schema $e$, $f$). We also compared setting the targets as individual objects (schema $c$, $e$) versus targets as entire scenes (schema $d$, $f$).

We visually inspect the efficacy of this self-supervised loss in encouraging better representations (beyond improvements of scores on tasks) in Appendix C.

## 3 EXPERIMENTS

We tested our model on two datasets, CLEVRER (Yi et al., 2020) and CATER (Girdhar & Ramanan, 2020). For each dataset, we pretrained a MONet model on individual frames. More training details and a table of hyperparameters are given in Appendix A.3.

### 3.1 CLEVRER

CLEVRER features videos of CLEVR objects (Johnson et al., 2016) that move and collide with each other; these objects are not necessarily visible in every frame. For each video, several questions are posed to test the model's understanding of what happened (or might happen). Unlike most other visual question answering datasets, which test for only descriptive understanding ("what happened in the video?"), CLEVRER explicitly poses other more causally-complex questions, including explanatory questions ("why did something happen?"), predictive questions ("what will happen next?"), and counterfactual questions ("what would happen in a different circumstance?") (Yi et al., 2020).

We compare our model to state of the art models reported in the literature: MAC (V+) and NS-DR from Yi et al. (2020), as well as the DCL model from Anonymous (2021a) (simultaneous to our work). MAC (V+) (based on the MAC network introduced in Hudson & Manning (2018)) is an end-to-end network that combines visual and language representations. It is augmented with object information and trained using ground truth labels for object segmentation masks and features (e.g. color, shape). NS-DR and DCL are hybrid models that apply a symbolic logic engine to outputs of various neural networks. The neural networks are used to detect objects, predict dynamics, and parse the question into a program, and the symbolic executor runs the parsed program to obtain the final output. NS-DR is trained using ground truth labels and ground truth parsed programs, while DCL requires only the ground truth parsed programs.

Table 1 shows the result of our model compared to these models; our model is also listed in the public leaderboard provided by Yi et al. (2020) under the name "neural"[1]. Across all categories, our model significantly outperforms the previous best models. Moreover, compared to the other models, our model does not use any labeled data other than the correct answer for the questions, nor does it require pretraining on any other dataset. Our model also was not specifically designed for this task, and it straightforwardly generalizes to other tasks as well, such as CATER (Girdhar & Ramanan, 2020).

**Detailed analysis**    Some sample model classifications on a randomly selected set of videos and questions are provided in Appendix D.1. These examples suggest qualitatively that, for most instances where the model was incorrect, humans would plausibly furnish the same answer. During further analysis of our results, we observed a spectrum of difficulties for CLEVRER's counterfactual questions, which we discuss in Appendix B. We find that our model scores 59.8% on the hardest counterfactual questions, which is still substantially better than both chance and all other models, albeit with some room for improvement remaining. Finally, to shed light on how our model arrives at its predictions, we provide detailed analysis of attention weights in Appendix C.

**Model ablation**    Table 1 shows the contributions of various components of our model. First, self-attention is necessary for solving this problem. For comparison, we replace our model's transformer with four fully connected layers with 2048 units per layer[2]. We find that an MLP is unable to answer non-descriptive questions effectively, despite using more parameters (20M vs 15M parameters).

Second, we verify that an object-based discretization scheme is essential to the performance of our model. We compare with a version of the architecture where the MONet object representations $\mu_{ti}$ are replaced with ResNet hyperpixels as in Zambaldi et al. (2019)[3]. Concretely, we flatten the output of the final convolutional layer of the ResNet to obtain a sequence of feature vectors that is fed into the transformer as the discrete entities. We find that an object level representation, such as one output by MONet, greatly outperforms the locality-aware but object-agnostic ResNet representation.

We also observe the importance of global attention between all objects across all frames, compared to a hierarchical attention model where objects within a frame could attend to each other but frames could only attend to each other as an atomic entity. We hypothesize that global attention may be important because with hierarchical attention, objects in different frames can only attend to each other at the "frame" granularity. A cube attending to a cube in a different frame would then gather information about the other non-cube objects, muddling the resulting representation. Since we care about how objects evolve over time, not operating at the level of objects is intuitively problematic.

Finally, we see that an auxiliary self-supervised (infill) loss improves the performance of the model by between 4 and 6 percentage points, with the greatest improvement on the counterfactual questions.

**Self-supervision strategies**    We compared the various masking schemes and loss functions for our auxiliary loss; a detailed figure is provided in Appendix A (Figure 4). We find that for all question types of the CLEVRER task, an L2 loss performs better than a contrastive loss, and among the masking schemes, masking one object per frame is the most effective. This particular result runs counter to our hypothesis that predictions or infilling in which the target is temporally removed from the context could encourage the model to learn more about scene dynamics and object interactions than (BERT-style) local predictions of adjacent targets. Of course, there may be other settings or loss functions that reveal the benefits of non-local prediction or constrastive losses; we leave this investigation to future work.

**Data efficiency**    We investigated how model performance varies as a function of the number of labelled (question-answer) pairs it learns from. To do so, we train models on $N\%$ of the videos and their associated labeled data. We evaluate the effect of including the auxiliary self-supervised loss

---

[1] https://evalai.cloudcv.org/web/challenges/challenge-page/667/leaderboard/1813

[2] We also tried a bidirectional LSTM, which achieved even lower performance. This may be because the structure of our inputs requires the learning of long-range dependencies.

[3] Note that in contrast to the ResNet-based models in Yi et al. (2020), our ResNet was not pretrained on ImageNet or any other dataset, but was simply trained with the rest of the model on the task.

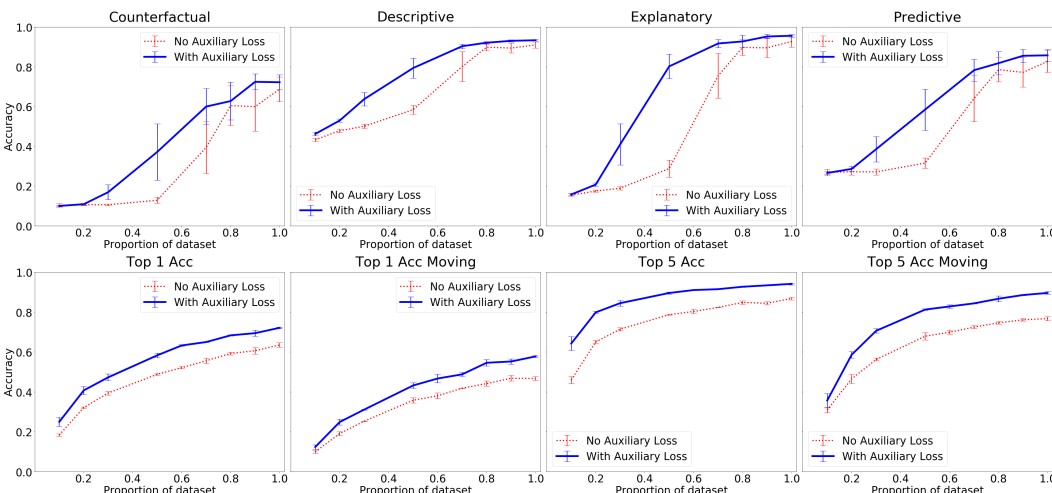

Figure 3: Accuracy with/without auxiliary loss for different proportions of CLEVRER (row 1) and CATER (row 2) training data.

(applied to the entire dataset, not just the labelled portion) in this low data regime. This scenario, where unlabeled data is plentiful while labeled data is scarce, occurs frequently in practice, since collecting labeled data is much more expensive than collecting unlabeled data.

Figure 3 shows that our best model reaches the approximate level of the previous state-of-the-art approaches using only 50%-60% of the data. We see that the self-supervised auxiliary loss makes a particular improvement to validation performance in low-data regimes. For instance, when trained on only 50% of the available labelled data, self-supervised learning enables the model to reach a performance of 37% on counterfactual questions (compared to 25% by MAC (V+) and 42% by NS-DR), while without self-supervised learning, the model only reaches a performance of 13% (compared to the 10% achieved by answering randomly (Yi et al., 2020)).

## 3.2 CATER

We also tested our model on CATER, an object-tracking dataset Girdhar & Ramanan (2020). In CATER, objects from the CLEVR dataset (Johnson et al., 2016) move and potentially cover other such objects, and the goal is to predict the location of a target object (called the *snitch*) in the final frame. Because the target object could be covered by multiple objects that could move in the meantime, the model must be sensitive to notions such as such as object permanence in order to track the target. There are two main variants of the CATER dataset, static camera and moving camera, differing in whether or not the camera that produces the video could move. A moving camera introduces additional complexity in that the model has to understand the camera motion and take that into account when making its prediction.

Table 2 shows our model compared to state of the art models in the literature on both static and moving camera videos. R3D is an implementation of I3D (Carreira & Zisserman, 2017) using ResNets in Wang et al. (2018), which also introduced the addition of non-local interactions; R3D and R3D non-local are the strongest two models evaluated by Girdhar & Ramanan (2020) OPNet, or the Object Permanence network (Shamsian et al., 2020), is an architecture with inductive biases designed for object tracking tasks; it was trained with extra supervised labels, namely the bounding boxes for all objects (including occluded ones). Hopper is a multi-hop transformer model from Anonymous (2021b) developed simultaneously with this work.

We train our model simultaneously on both static and moving camera videos. Our model outperforms the R3D models for both static and moving cameras. We also ran our model with an additional auxiliary loss consisting of the L1 distance between the predicted cell and the actual cell. With this additional loss, we get comparable results in the *moving* camera case as the R3D models for the *static* camera case. Moreover, we achieve comparable accuracy as OPNet for accuracy and L1 distance, despite requiring less supervision to train. Appendix D.2 gives a few sample outputs of our model;

| Model | Top 1 (S) | Top 5 (S) | L1 (S) | Top 1 (M) | Top 5 (M) | L1 (M) |
|---|---|---|---|---|---|---|
| R3D LSTM | 60.2 | 81.8 | 1.2 | 28.6 | 63.3 | 1.7 |
| R3D + NL LSTM | 46.2 | 69.9 | 1.5 | 38.6 | 70.2 | 1.5 |
| OPNet | **74.8** | - | 0.54 | - | - | - |
| Hopper | 73.2 | 93.8 | 0.85 | - | - | - |
| Ours (no auxiliary) | 60.5 | 84.5 | 0.90 | 46.8 | 75.1 | 1.3 |
| Ours | 70.6 | 93.0 | 0.53 | 56.6 | 87.0 | 0.82 |
| Ours (with L1 loss) | $74.0 \pm 0.3$ | $\mathbf{94.0 \pm 0.4}$ | $\mathbf{0.44 \pm 0.01}$ | $\mathbf{59.7 \pm 0.5}$ | $\mathbf{90.1 \pm 0.6}$ | $\mathbf{0.69 \pm 0.01}$ |

Table 2: Performance on CATER of our model compared to the best results from literature. We report top 1 accuracy, top 5 accuracy, and L1 distance between the predicted grid cell and true grid cell. The labels (S) and (M) refer to static and moving cameras.

in particular we note that it is able to find the target object in several cases where the object was occluded, demonstrating that our model is able to do some level of object tracking. Finally,we find that an auxiliary self-supervised loss helps the model perform well in the low data regime for CATER as well, as shown in Figure 3.

## 4 RELATED WORK

**Self-attention for reasoning** Various studies have shown that transformers (Vaswani et al., 2017) can manipulate symbolic data in a manner traditionally associated with symbolic computation. For example, in Lample & Charton (2020), a transformer model learned to do symbolic integration and solve ordinary differential equations symbolically, tasks traditionally reserved for symbolic computer algebra systems. Similarly, in Hahn et al. (2020), a transformer model learned to solve formulas in propositional logic and demonstrated some degree of generalization to out of distribution formulas. Finally, Brown et al. (2020) showed that a transformer trained for language modeling can also do simple analogical reasoning tasks without explicit training. Although these models do not necessarily beat carefully tuned symbolic algorithms (especially on out of distribution data), they are an important motivation for our proposed recipe for attaining strong reasoning capabilities from self-attention-based models on visually grounded tasks.

**Object representations** A wide body of research points to the importance of object segmentation and representation learning. Various supervised and unsupervised methods have been proposed for object detection and feature extraction (Ren et al., 2015; He et al., 2017; Burgess et al., 2019; Greff et al., 2019; Lin et al., 2020; Du et al., 2020). Past research have also investigated using object based representations in downstream tasks (Raposo et al., 2017; Desta et al., 2018).

**Self-supervised learning** Another line of research concerns learning good representations through self-supervised learning, with an unsupervised auxiliary loss to encourage the discovery of better representations. These better representations could lead to improved performance on supervised tasks, especially when labeled data is scarce. In Devlin et al. (2018), for instance, an auxiliary infill loss allows the BERT model to benefit from pretraining on a large corpus of unlabeled data. Our approach to object-centric self-supervised learning is heavily inspired by the BERT infilling loss. Other studies have shown similar benefits to auxiliary learning in the vision domain as well (Gregor et al., 2019; Han et al., 2019; Chen et al., 2020). These works apply various forms of contrastive losses to predict scene dynamics. The better representations that these contrastive losses encourage carry downstream benefits to supervised and reinforcement learning tasks.

**Vision and language in self-attention models** Recently, many works have emerged on applying transformer models to visual and multimodal data, for static images (Li et al., 2019; Lu et al., 2019; Tan & Bansal, 2019; Su et al., 2020) and videos (Zambaldi et al., 2019; Sun et al., 2019b;a). These approaches combine the output of convolutional networks with language in various ways using self-attention. While these previous works focused on popular visual question answering tasks, which typically consist of descriptive questions (Yi et al., 2020), we focus on understanding deeper causal dynamics of videos. Together with these works, we provide more evidence that self-attention between visual and language elements enables good performance on a diverse set of tasks. In addition, while

the use of object representations for discretization in tasks involving static images is becoming more popular, the right way to discretize videos is less clear. We provide strong evidence in the form of ablation studies for architectural decisions that we claim are essential for higher reasoning for this type of data: visual elements should correspond to physical objects in the videos and inter-frame attention between sub-frame entities (as opposed to inter-frame attention of entire frames) is crucial. We also demonstrate the success of using unsupervised object segmentation methods as opposed to the supervised methods used in past work.

## 5 CONCLUSION

We apply a self-attention model to videos discretized into objects and show that such a model is able to understand video dynamics and perform causal reasoning. Our model substantially outperforms all previous state of the art methods on all metrics for two different datasets, including hybrid neuro-symbolic architectures hand-coded and explicitly designed to solve a specific task. This result adds to the growing body of evidence that neural networks, particularly self-attention architectures, could do reasoning tasks that traditionally only symbolic logic AI excel at (Lample & Charton, 2020; Mitchell, 2020). We also show that discretization at the object level is essential to the performance of our model. Finally, we demonstrate that many techniques from natural language processing, such as the classification token and masking/infilling, apply in the visual domain as well. We hope that this bridge between video understanding research and natural language research will facilitate idea sharing between the domains.

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

# A    METHODS DETAILS

## A.1    MONET

To segment each $w \times h$ frame $F_t$ into $N_o$ object representations, MONet uses a recurrent attention network to obtain $N_o$ attention masks $\mathbf{A}_{ti} \in [0,1]^{w \times h}$ for $i = 1, \ldots, N_o$ that represent the probability of each pixel in $F_t$ belonging to the $i$-th object, with $\sum_{i=1}^{N_o} \mathbf{A}_{ti} = 1$. This attention network is coupled with a component VAE with latents $\mathbf{z}_{ti} \in \mathbb{R}^d$ for $i = 1, \ldots, N_o$ that reconstructs $\mathbf{A}_{ti} \odot F_t$, the $i$-th object in the image. The latent posterior distribution $q(\mathbf{z}_t | F_t, \mathbf{A}_{ti})$ is a diagonal Gaussian with mean $\mu_{ti}$, and we use $\mu_{ti}$ as the representation of the $i$-th object.

## A.2    SELF-SUPERVISED TRAINING

Recall in the main text that we wrote the auxiliary self-supervised loss as

$$\text{auxiliary loss} = \sum_{t,i} \tau_{ti} l\left(f(\mu'_{ti}), \mu\right).$$

We tested an L2 loss and a contrastive loss (inspired by the loss used in Han et al. (2019)), and the formulas for the two losses are respectively:

$$l_{\text{L2}}\left(f(\mu'_{ti}), \mu\right) = \|f(\mu'_{ti}) - \mu_{ti}\|_2^2$$
$$l_{\text{contrastive}}\left(f(\mu'_{ti}), \mu\right) = -\log \frac{\exp(f(\mu'_{ti}) \cdot \mu_{ti})}{\sum_{s,j} \exp\left(f(\mu'_{ti}) \cdot \mu_{sj}\right)}.$$

A comparison of these losses and the masking schemes is given in Figure 4.

We also tested a few variations of the contrastive loss inspired by literature and tested all combinations of variations. The first variation is where the negative examples all come from the same frame:

$$l_{\text{contrastive}}\left(f(\mu'_{ti}), \mu\right) = -\log \frac{\exp(f(\mu'_{ti}) \cdot \mu_{ti})}{\sum_j \exp\left(f(\mu'_{ti}) \cdot \mu_{tj}\right)}.$$

The second variation is adding a temperature $\tau$ to the softmax (Chen et al., 2020):

$$l_{\text{contrastive}}\left(f(\mu'_{ti}), \mu\right) = -\log \frac{\exp(f(\mu'_{ti}) \cdot \mu_{ti})/\tau}{\sum_{s,j} \exp\left(f(\mu'_{ti}) \cdot \mu_{sj}/\tau\right)}.$$

The final variation we tested is using cosine similarity instead of dot product:

$$l_{\text{contrastive}}\left(f(\mu'_{ti}), \mu\right) = -\log \frac{\exp(\text{sim}(f(\mu'_{ti}), \mu_{ti}))}{\sum_{s,j} \exp\left(\text{sim}(f(\mu'_{ti}), \mu_{sj})\right)}.$$

where $\text{sim}(\mathbf{x}, \mathbf{y}) = \frac{\mathbf{x} \cdot \mathbf{y}}{\|\mathbf{x}\| \cdot \|\mathbf{y}\|}$. We found that these variations did not significantly change the performance of the model (and the optimal temperature setting was close to $\tau = 1$), and leave to future work more careful analysis of these contrastive losses and the representations they encourage.

## A.3    TRAINING DETAILS

We generally follow similar training procedures as for the models described in Yi et al. (2020) and Girdhar & Ramanan (2020).

For CLEVRER, we resize videos to 64 by 64 resolution and sample 25 random frames, as in Yi et al. (2020). We divide our batch of 16 videos and questions in two, a supervised sub-batch and an unsupervised sub-batch (where question answers are not provided). The supervised sub-batch is used to calculate the classification loss, and the unsupervised sub-batch is used to calculate the unsupervised auxiliary loss. This division was made so that we can use a subset of available data for the supervised sub-batch while using all data for the unsupervised sub-batch. The supervised sub-batch is further subdivided into two sub-batches of size 4, for descriptive and multiple choice questions (this division was made since the output format is different for the two types of questions).

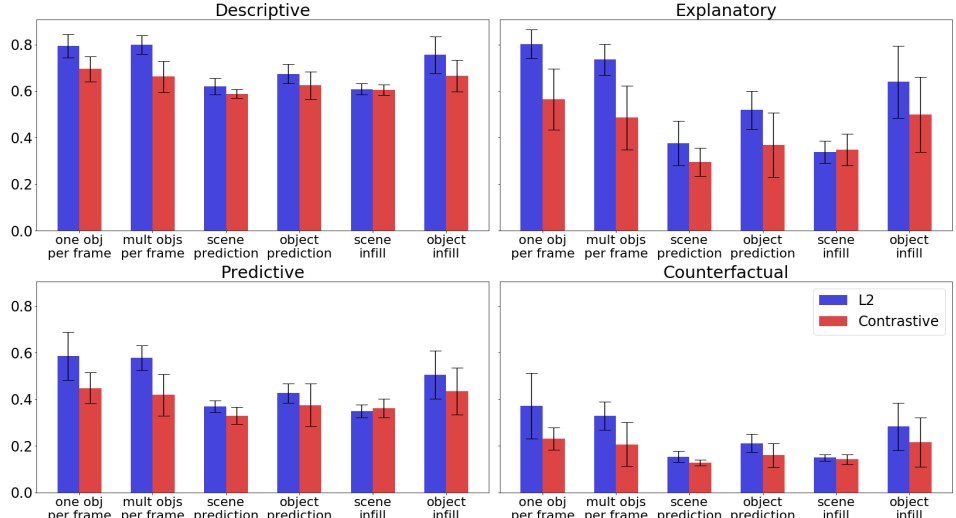

Figure 4: Comparison of different mask types and loss functions for auxiliary loss computation. Models were trained on 50% of the CLEVRER dataset to magnify the effects of the self-supervised loss.

For CATER, we also resize videos to 64 by 64 resolution and sample 80 random frames. We train on static and moving camera data simultaneously, with the batch of 8 videos divided equally between the two.

For both datasets, we pretrain a MONet model on frames extracted from the respective dataset. The training of the MONet models follow the procedures described in Burgess et al. (2019).

Motivated by findings from language modeling, we trained the main transformer model using the LAMB optimizer (You et al., 2019) and found that it offered a significant performance boost over the ADAM optimizer (Kingma & Ba, 2014) for the CLEVRER dataset (data not shown). Results converge after 200,000 steps for CLEVRER and 60,000 steps for CATER. All error bars are computed over at least 5 seeds. The below table lists the hyperparameters used in our model.

| Parameter | Value | Parameter | Value |
|---|---|---|---|
| Batch-size | 16 | Batch-size | 8 |
| Transformer heads | 10 | Transformer heads | 8 |
| Transformer layers | 28 | Transformer layers | 16 |
| Embedding size $d$ | 16 | Embedding size $d$ | 36 |
| Number of objects $N_o$ | 8 | Number of objects $N_o$ | 8 |
| Prediction head hidden layer size | 128 | Prediction head hidden layer size | 144 |
| Learning rate | 0.002 | Learning rate | 0.002 |
| Learning rate warmup steps | 4000 | Learning rate warmup steps | 4000 |
| Infill cost $\lambda$ | 0.01 | Infill cost $\lambda$ | 2.0 |

(a) Hyperparameters for CLEVRER.      (b) Hyperparameters for CATER.

## B   ANALYSIS OF CLEVRER DATASET

During analysis of our results, we noticed that some counterfactual questions in the CLEVRER dataset can be solved without using counterfactual reasoning. In particular, about 47% of the counterfactual questions ask about the effect of removing an object that did not collide with any other object, hence having no effect on object dynamics; an example is given in Figure 5. Moreover, even for the questions where the removed object is causally connected to the other objects, about 45% can be answered perfectly by an algorithm answering the question as if it were a descriptive question. To quantify this, we wrote a symbolic executor that uses the provided ground-truth video annotations

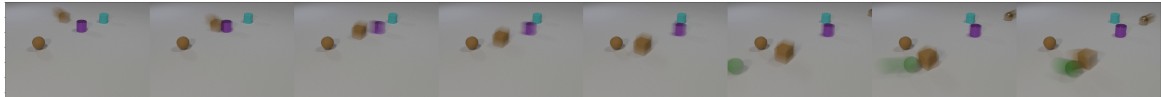

Figure 5: The video for an example counterfactual question that can be answered as if it were a descriptive question. The question is: if the brown rubber sphere is removed, what will not happen?

and parsed questions to determine causal connectivity and whether each choice happened in the non-counterfactual scenario.

Although determining whether or not a given counterfactual question can be answered this way still requires counterfactual reasoning, we want to eliminate the possibility that our model achieved its 75% accuracy on counterfactual questions without learning counterfactual reasoning; instead it might have reached that score simply by answering all counterfactual questions as descriptive questions. To verify this is not the case, we evaluated our model on only the harder category of counterfactual questions where the removed object does collide with other objects and which cannot be answered by a descriptive algorithm. We find that our model achieves a performance of 59.8% on this harder category. This is significantly above chance, suggesting that our model is indeed able to do some amount of true counterfactual reasoning.

## C   QUALITATIVE ANALYSIS

We provide qualitative analysis of attention weights in order to shed light on how our model arrives at its predictions. These examples illustrate broad patterns evident from informal observation of the model's attention weights. We focus on the following video from CLEVRER:

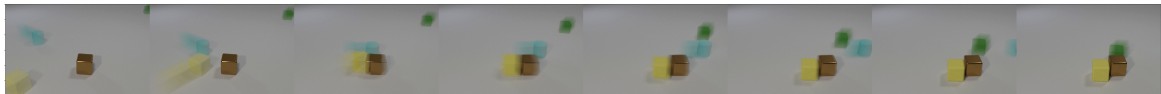

In this video, a yellow rubber cube collides with a cyan rubber cylinder. The yellow cube then collides with a brown metallic cube, while the cyan cylinder and a green rubber cube approach each other but do not collide. Finally, the green cube approaches but does not collide with the brown cube.

**Cross-modal attention**   We analyzed the cross-modal attention between words in the question and the MONet objects in one frame of the video. For each word, we determined the MONet object that attended to that word the most. In particular, we looked at the attention weights in the last layer of the transformer for one head (of the multi-head attention). The result is shown in the visualization below. We drew bounding boxes corresponding to the objects as determined by MONet, and we colored each word in the question according to the MONet object that attended to that word with highest weight (black represents a MONet slot without any objects). We observe that, for this head, objects attend heavily to the words that describe them. For example the cyan cylinder attends heavily to the words "cylinder" and "removed".

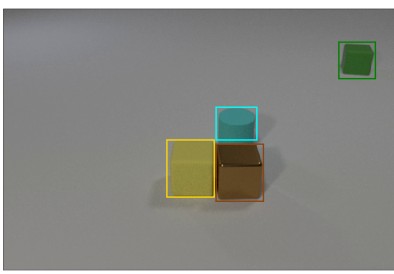

**Q:** If the cylinder is removed, which event will not happen?

1. The brown object collides with the green object.

2. The yellow object and the metal cube collide.

3. The yellow cube collides with the green object.

**Most important objects**   For each frame of the video, we look at the objects that were most heavily attended upon in determining the final answer. We measure the attention weights (for one head) in the last layer of the transformer for the $CLS$ token attending on each object. The image below

illustrates the results, when the model is tasked with assessing the likelihood of the first choice of the above counterfactual question (whether the green object will collide with the brown object if the cyan cylinder is removed). We drew bounding boxes for the two most important objects of each frame, as determined by the attention weights. Note that our MONet was configured to detect eight objects (including the background); the transformer did not select the "empty" object slots nor the background slot. We observe that this head generally focuses on the green and brown objects, but switches its focus to the cyan cylinder and brown cube when it looks like the cylinder's interaction with the yellow cube could potentially change the outcome.

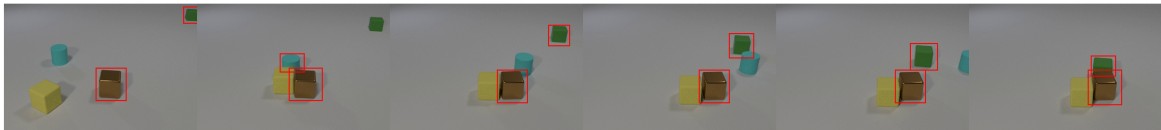

The relative importance of the various objects depends on the question the model is answering. When the model is tasked with a predictive question (whether or not the cylinder and the green cube will collide) a different attention pattern emerges. Here, we observe one head of the transformer focusing on collisions: first the collision of the cylinder and the yellow cube, then on the cylinder and the green cube when they move towards each other.

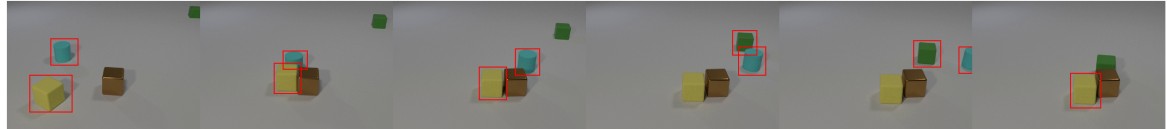

**Object alignment** Recall that MONet does not assign objects to slots in a well-determined manner— tiny changes in an image can cause MONet to unpredictably assign objects to slots in a different permutation. The transformer is able to maintain object identity when MONet outputs objects in a different order. The image below, where we again show the two most attended-upon objects for each frame, illustrate instances where MONet changes the permutation of objects. In this image, we plot time on the x-axis and MONet slot index on the y-axis; the slots containing the two most important objects are grayed out. We observe that the transformer is able to align objects across time, maintaining consistent attention to the green and brown objects.

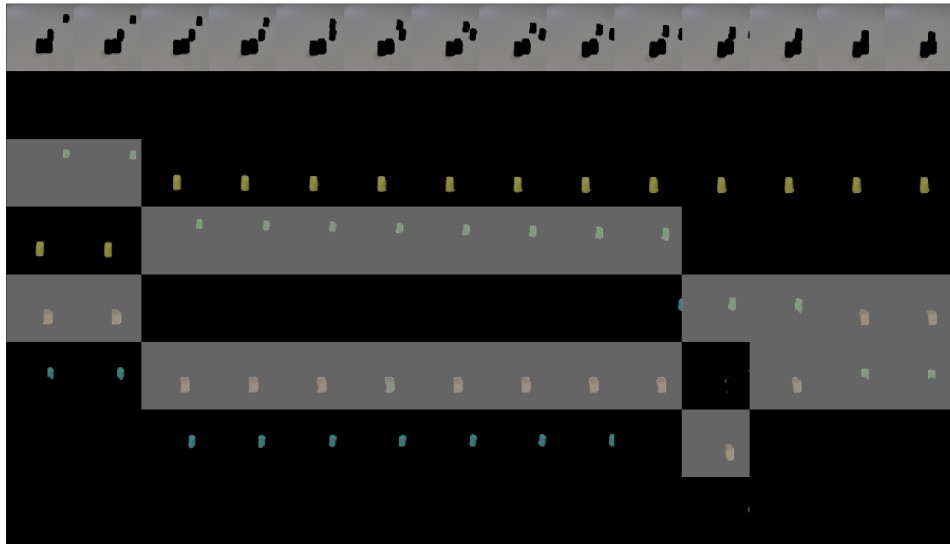

**Effectiveness of the auxiliary loss** Finally, we visually inspect our hypothesis that our self-supervised loss encourages the transformer in learning better representations. For clarity of the subsequent illustration, we use the scene prediction masking scheme, as described in Figure 2. In this scheme, the transformer has to predict the contents of the last few frames (the *target frames*) given the beginning of the video. To pose harder predictive challenges, we mask out the three frames preceding

the target frames in addition to the target frames themselves. The two images below compare the predicted frames (second image) to the true frames (first image). In the second image, the black frames are the three masked out frames preceding the target frames. The frames following the black frames are the target frames; they contain the MONet-reconstructed images obtained from latents predicted by the transformer. The frames preceding the black frames are MONet-reconstructed images obtained from the original latents (the latents input into the transformer).

We observe that with the self-supervised loss, we get coherent images from the transformer-predicted latents with all the right objects (in the absence of the auxiliary loss, the transformed latents generate incoherent rainbow blobs). We also observe the rudiments of prediction, as seen in the movement of the yellow object in the predicted image. Nevertheless, it is also clear that the transformer's predictions are not perfect, and we leave improvements of this predictive infilling to future work.

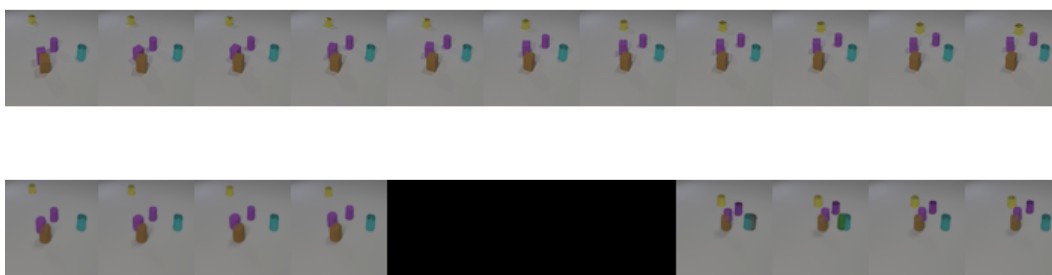

# D  EXAMPLE MODEL PREDICTIONS

In this section, we provide a few sample classifications produced by our model. All examples are produced at random from the validation set; in particular we did not cherry-pick any examples to highlight the performance of our model.

## D.1  CLEVRER

We provide four videos and up to two questions per question type for the video (many videos in the dataset come with only one explanatory or predictive question). For each question type with more than one question, we try to choose one correct classification and one misclassification if available to provide for greater diversity. Besides this editorial choice, all classifications are sampled randomly.

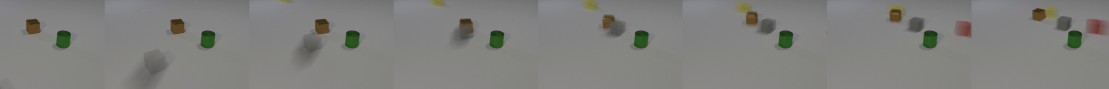

**Q:** How many metal objects are moving?
**Model:** 1
**Label:** 1

**Q:** What is the shape of the stationary metal object when the red cube enters the scene?
**Model:** cylinder
**Label:** cylinder

**Q:** Which of the following is not responsible for the collision between the metal cube and the yellow cube?

1. the presence of the gray cube
2. the gray object's entrance
3. the presence of the red rubber cube
4. the collision between the gray cube and the metal cube

**Model:** 3
**Label:** 3

**Q:** Which event will happen next?

1. The gray object collides with the red object
2. The gray object and the cylinder collide

**Model:** 1
**Label:** 1

**Q:** Which event will happen if the red object is removed?

1. The gray object and the brown object collide
2. The gray object collides with the cylinder
3. The gray cube collides with the yellow object
4. The brown cube and the yellow object collide

**Model:** 1, 4
**Label:** 1, 4

**Q:** What will happen if the cylinder is removed?

1. The brown cube collides with the red cube
2. The red object and the yellow object collide
3. The gray cube collides with the red cube
4. The gray object collides with the brown object

**Model:** 3, 4
**Label:** 3, 4

**Q:** What color is the metal object that is stationary when the metal cube enters the scene?
**Model:** blue
**Label:** blue

**Q:** What material is the last object that enters the scene?
**Model:** metal
**Label:** rubber

**Q:** Which of the following is not responsible for the collision between the cyan object and the sphere?

1. the presence of the red rubber object
2. the red object's entering the scene
3. the collision between the sphere and the blue cube

**Model:** 1, 2, 3
**Label:** 1, 2, 3

**Q:** What will happen next?

1. The metal cube and the red cube collide
2. The sphere collides with the metal cube

**Model:** 1
**Label:** 1

**Q:** Without the red cube, which event will happen?

1. The sphere collides with the blue cube
2. The cyan object and the blue cube collide

**Model:** 1
**Label:** 1

**Q:** What will not happen without the sphere?

1. The cyan object collides with the red cube
2. The cyan object collides with the metal cube
3. The metal cube and the red cube collide

**Model:** 1, 3
**Label:** 3

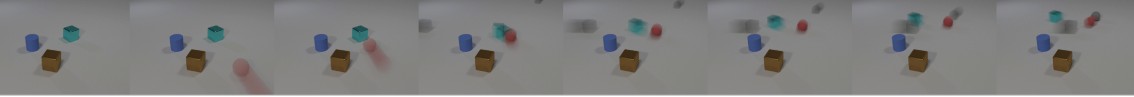

**Q:** Are there any moving brown objects when the red object enters the scene?
**Model:** no
**Label:** no

**Q:** How many rubber objects are moving?
**Model:** 3
**Label:** 3

**Q:** Which of the following is not responsible for the collision between the red object and the gray sphere?

1. the presence of the gray cube
2. the collision between the red object and the cyan object
3. the rubber cube's entering the scene
4. the presence of the cyan object

**Model:** 1, 3
**Label:** 1, 3

**Q:** What will happen next?

1. The gray cube and the brown object collide
2. The red object collides with the rubber cube

**Model:** 2
**Label:** 2

**Q:** If the cylinder is removed, which of the following will not happen?

1. The gray cube and the brown cube collide
2. The red object and the cyan object collide
3. The red sphere and the rubber cube collide
4. The cyan object and the brown cube collide

**Model:** 1, 4
**Label:** 1, 4

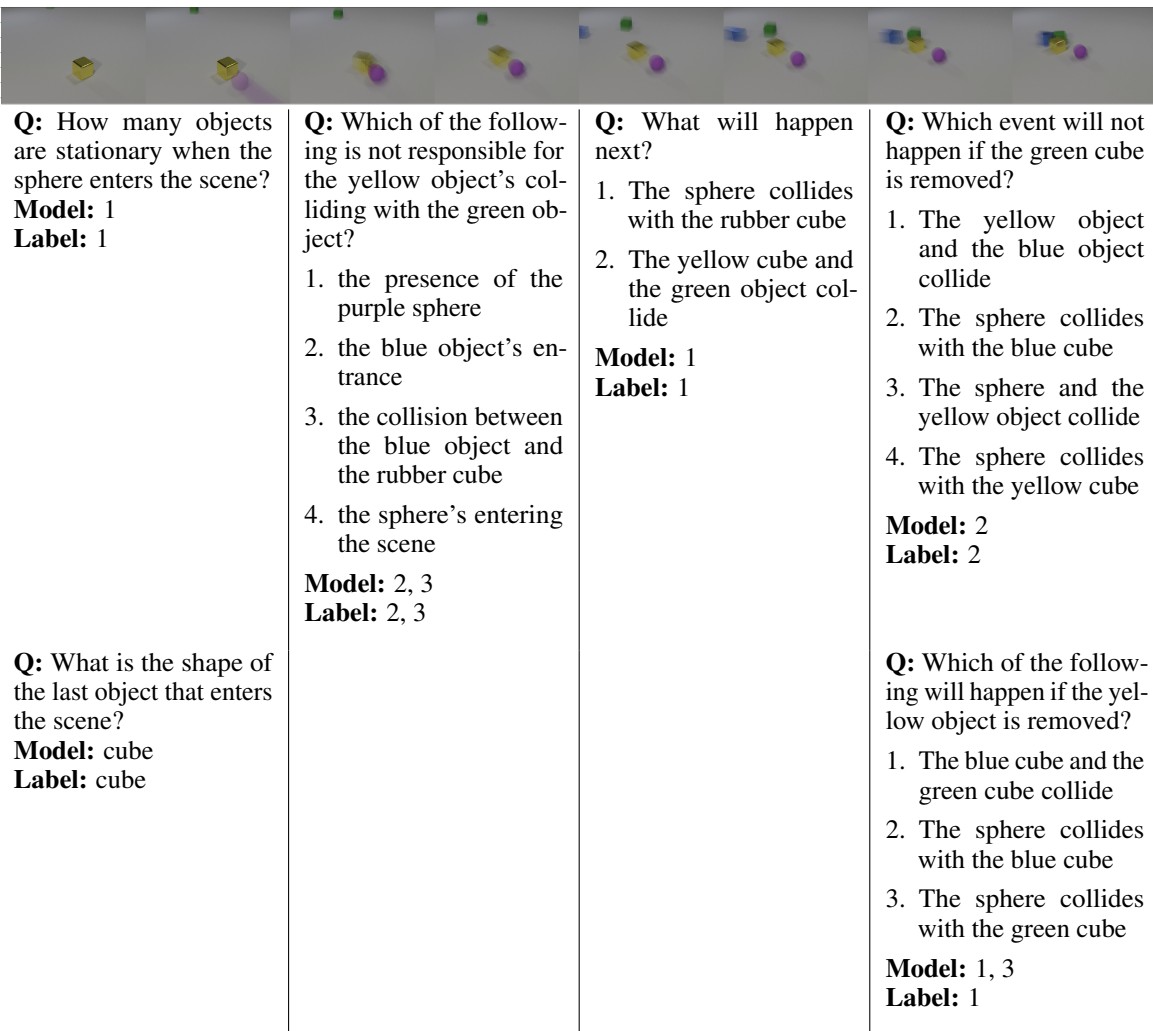

**Q:** How many objects are stationary when the sphere enters the scene?
**Model:** 1
**Label:** 1

**Q:** What is the shape of the last object that enters the scene?
**Model:** cube
**Label:** cube

**Q:** Which of the following is not responsible for the yellow object's colliding with the green object?

1. the presence of the purple sphere
2. the blue object's entrance
3. the collision between the blue object and the rubber cube
4. the sphere's entering the scene

**Model:** 2, 3
**Label:** 2, 3

**Q:** What will happen next?

1. The sphere collides with the rubber cube
2. The yellow cube and the green object collide

**Model:** 1
**Label:** 1

**Q:** Which event will not happen if the green cube is removed?

1. The yellow object and the blue object collide
2. The sphere collides with the blue cube
3. The sphere and the yellow object collide
4. The sphere collides with the yellow cube

**Model:** 2
**Label:** 2

**Q:** Which of the following will happen if the yellow object is removed?

1. The blue cube and the green cube collide
2. The sphere collides with the blue cube
3. The sphere collides with the green cube

**Model:** 1, 3
**Label:** 1

## D.2 CATER

We include ten random videos from the validation subset of the static camera CATER dataset. In the final frame of the video, the correct grid cell of the target snitch is drawn in blue, and the model's prediction is drawn in red. We note that the model is able to find the snitch in scenarios where the snitch is hidden under a cone that later moves (along with the still hidden snitch); in the sixth example, the model also handled a case where the snitch was hidden under two cones at some point in time.

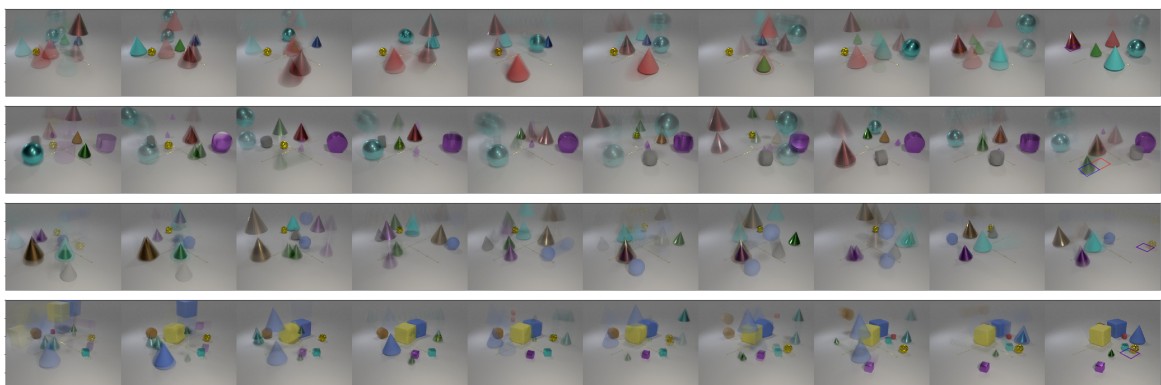

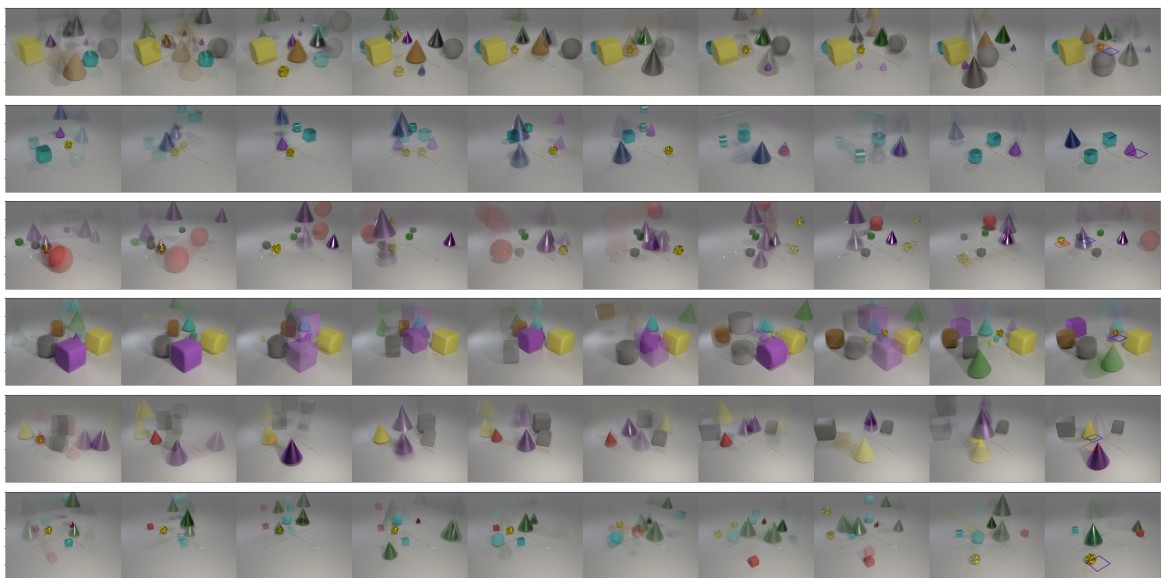

