# OpenReview forum: "Neural spatio-temporal reasoning with object-centric self-supervised learning"
_ICLR.cc/2021/Conference — Reject_

### Official Review · AnonReviewer2 · 2020-10-27
**Results on performance and data efficiency are impressive, but how should we interpret such results? I think the results presented here are too premature to support the claimed contribution.**

**Rating:** 5
**Confidence:** 4

**Review:**

This paper proposes to use a self-attention component in combination with unsupervised object representation learning (e.g., MONet) for CLEVRER and CATER. Results show considerable performance improvement and data efficiency compared with prior methods.

Pros:

+ This paper is easy to follow.

+ Results on performance and data efficiency are impressive.

Cons:

- It is a common misconception that performance is still the first priority in artificial tasks like CLEVRER or CATER. It is not. Artificial tasks are designed to probe specific capabilities. Hence, it is vital to show that a model indeed learns these capabilities before drawing any firm conclusions; for instance, the model is capable of counterfactual and causal reasoning. The authors fail to justify this aspect in the current version.

- Specifically, although it is natural and reasonable to follow prior evaluation metrics, it is essential to ask whether the prior setting is still meaningful in the new context. This paper is set to test whether self-attention-based neural networks are able to perform causal reasoning and physical understanding. The important context here is that such a family of neural networks has been very successful in more complex tasks, hence given an artificial task and smaller size of training data, it is predictive that the model would perform better. Note that this is not new to visual inputs; for instance, the Named Detection Transformer (DETR) also uses similar components for segmentation. Also note that whether such a family of model indeed possesses a certain level of reasoning is still debatable, as some researchers found in conversational reasoning (e.g., [1,2]) and in particular, advocated by Gary Marcus. Given these contexts, it is natural to ask whether the proposed method, in fact, possesses a certain level of reasoning capability instead of merely fitting the data.

- To refute such a potential hypothesis, what could we do? Since the goal is to test whether such a model is indeed capable of causal reasoning or physical understanding, I think the authors need to present something beyond the prior evaluation metrics to really justify their claims. Otherwise, we are simply far more premature to draw such a conclusion that the proposed model, built on top of existing knowledge of neural networks, is capable of causal reasoning and physical understanding.

- To verify a learned model is capable of physical understanding, can we train the model on one task domain and test its generalization in another task domain? Physical understanding should be very transferrable once learned, akin to a physical engine that simulates the virtual world; one does not need tens of engines, and one suffices. Similarly, cognition has presented tons of evidence [3], showing how humans/infants react to causal reasoning tasks. Can we augment the existing dataset to show such a capability? In this way, we can have sufficient pieces of evidence to support the claimed contributions.

- In summary, adopting a powerful and newer model on existing artificial tasks, one needs to be careful to draw conclusions by presenting sufficient pieces of evidence beyond a single measurement of performance. The current experimental results, in my opinion, are not sufficient to support the claimed contributions.

- Some sections of the writing are more than necessary. For instance, it is completely unnecessary to spend almost 2 pages on various sections talking about self-attention in Transformer. Most of these backgrounds and information should be assumed known to a reviewer at the top conference.

[1] From Eliza to XiaoIce: challenges and opportunities with social chatbots
[2] Augmenting end-to-end dialog systems with commonsense knowledge
[3] A theory of causal learning in children: Causal maps and Bayes nets

---

> ### Author Response · Authors · 2020-11-16
> **Re: Significance of results**
>
> Thanks for your assessment of our paper. We respectfully disagree that performance on these artificial tasks is irrelevant, and will try to convince you why! It is certainly true that no tasks and metrics are perfect; all tasks and metrics that try to measure a model’s ability to do something (image recognition, language translation, causal reasoning) only measure a proxy of that ability. Nevertheless, optimizing these proxies is still very useful (e.g., BLEU in language generation), not only because measuring the “true” ability is usually difficult, time intensive, and expensive, but also because proxy performance often correlates with “true” performance. We do not claim that solving CLEVRER and CATER implies solving causal reasoning, but we should also not deny that it is progress.
>
> We’d also like to be careful to not shift the goalposts; just a month ago these tasks were accepted as important, challenging probes of causal reasoning that bested top models to date. If one could develop a model that could do well *even on these proxy tasks*, it would be an advancement of the field. We also note that in the papers where these datasets were introduced [1, 2], it was claimed (and demonstrated) that this data specifically challenged the type of models presented in our work. We therefore don’t think it’s accurate to claim that “it is predictive that [our] model would perform better” when previous (and current [3]) works suggest otherwise; indeed, the demonstrated failures of these models were used to explicitly motivate a “paradigm shift” [4] towards neuro-symbolic methods.
>
> You ask “whether the proposed method, in fact, possesses a certain level of reasoning capability instead of merely fitting the data”. Without getting too philosophical, how would one demonstrate a capacity for causal reasoning if not by performing on tasks involving causal reasoning? Demonstrating generalization to more challenging tasks would of course improve our paper; this is why we included CATER as an orthogonal measurement of our model’s capabilities (we note that previous and current models under consideration have not shown this level of generalization [1, 2, 3, 5]). However, additional demonstrations would not and should not invalidate the success of our models on the tasks we studied, which were already seen as important challenges in the field.
>
> It is of course possible that our model would not do well on a more complex task of the same flavor; CLEVRER was designed as such a task for previous models, and we look forward to the development of new challenges to advance machine learning research. But until such a new challenge is designed, CLEVRER is, in our opinion, a suitable task for investigating causal reasoning. We think it would be unreasonable to require every paper showing modeling advancements to propose new and harder challenges, especially when these advancements are more than just incremental.
>
> If you believe that we have overstated our claims, then we are happy to correct them. We look forward to engaging with you further on this matter, and would appreciate specific pointers in our text.
>
> [1] K. Yi et al. CLEVRER: CoLlision Events for Video REpresentation and Reasoning. ICLR
> 2020
>
> [2] R. Girdhar and D. Ramanan. CATER: A diagnostic dataset for Compositional Actions and TEmporal Reasoning. ICLR 2020
>
> [3] Grounding Physical Object and Event Concepts Through Dynamic Visual Reasoning. Under review for ICLR 2021. https://openreview.net/forum?id=bhCDO_cEGCz
>
> [4] https://mitibmwatsonailab.mit.edu/research/blog/clevrer-the-first-video-dataset-for-neuro-symbolic-reasoning/
>
> [5] Hopper: Multi-hop Transformer for Spatiotemporal Reasoning. Under review for ICLR 2021. https://openreview.net/forum?id=MaZFq7bJif7

---

### Official Review · AnonReviewer3 · 2020-10-28

**Rating:** 5
**Confidence:** 4

**Review:**

This paper studies the temporal and spatial reasoning in videos. Specifically, the authors propose to combine unsupervised object representation learning MONet with self-attention transformer and introduce self-supervised learning through masked representation prediction. Experiments are conducted on CLEVRER and CATER dataset and the performance improvements compared to baselines validate the proposed methods.

Overall I think the idea in this paper is interesting, especially using unsupervised way to extract object-centric representation then using self-attention to learn the higher-level reasoning. The experiment results seem promising. However the idea of combining visual-linguistic features in transformer and pretraining with masked representation has been studied in previous works like VL-BERT: Pre-training of Generic Visual-Linguistic Representations and Learning Video Representations using Contrastive Bidirectional Transformer. So the novelty is a major concern. Also the effectiveness of temporal reasoning is degraded by the experimental results that "masking one object per frame is the most effective".

Another concern is some missing/vague technical details making the reading rather difficult, for example the section for different masking scheme.
Some detailed questions like:
Is representations learned from MONet learnable during masked representation prediction?
Is there a pretraining phase for the masked object representation prediction? Or the auxiliary loss is applied together with the classification loss?
Any reasons why not compare with other baselines in CATER like "Learning Object Permanence from Video" by Shamsian et al?
What does it mean by " MONet does not assign objects to slot indices in a well defined way"?
What's the attention design for "hierarchical attention model"? Is the representation input to transformer for the whole image rather than single object?

--------
After discussion:

After reading the author's reply as well as the opinions from other reviewers, I will stick to my original rating since 1) the writing makes the paper hard to understand 2) current experiments cannot support the central claim of spatial-temporal reasoning. While the author resolve some of the concerns, they are encouraged to further polish the paper and use more evidence to support their claims.

---

> ### Author Response · Authors · 2020-11-12
> **Re: Novelty and clarifications**
>
> Thank you for your review. Answers to the detailed questions you raise will surely improve the readability of the text, and we will address them at the end of this message.
>
> First, though, we’d like to respond to novelty being a “major concern”. The novelty in our work lies not in the use of an object-centric multi-modal transformer, but in its application to video-based question answering that involves causal reasoning. We’d like to emphasize some of the hypotheses underlying CLEVRER:
> * The CLEVRER paper [1] suggests that a dedicated dynamics model is required for causal reasoning on this data. We show this to be false.
> * The CLEVRER paper suggests that state of the art neural networks “lack the ability to perform causal reasoning and struggle on the explanatory, predictive, and counterfactual questions”. We show this to be false.
> * The CLEVRER authors present their neuro-symbolic model as part of a “neuro-symbolic paradigm shift” [2] that addresses the shortcomings of purely neural approaches. We demonstrate that these shortcomings are an illusion.
>
> Thus, the novelty of our work therefore primarily lies in the knowledge gained from our experiments, specifically in regards to hypotheses that were explicitly laid out in both published research and papers under consideration [1, 3].
>
> This aside, while it is true that our paper combined many pre-existing components, it did so in a principled way; our approach worked, as compared to the analogous baselines in the original papers. And that it did so without staking claim in yet another architecture should be seen as a positive feature.
>
> ------
> **Clarifications:**
> 1. "the effectiveness of temporal reasoning is degraded by the experimental results that 'masking one object per frame is the most effective'."
>
> We expected masking one object per frame to be the most effective solution due to the alignment problem: MONet can assign objects to slots in many different permutations. Tiny differences in the input image or MONet parameters could lead to MONet outputting the same objects but in different order. Therefore, even if a transformer successfully predicted the attributes of the missing objects, the transformer cannot know for certain which predicted object to assign to each masked-out slot; many permutations could have been output by MONet. This would lead to the model being penalized by the loss function even if it was essentially correct. To avoid this ambiguity, we tried masking only one slot per frame, and we indeed found this to work better.
>
> We do not understand your point about why this detracts from the model’s “temporal reasoning” ability. Could you clarify?
>
> 2. "Any reasons why not compare with other baselines in CATER like 'Learning Object Permanence from Video' by Shamsian et al"
>
> Thank you for bringing this paper to our attention; we will include it in our baselines. We note that in comparison to Shamsian et al, our model was trained with less supervision but nevertheless achieved almost the same performance (74% [ours] vs 74.8% [theirs, better] for top 1 accuracy, and 0.44 [ours, better] vs 0.54 [theirs] for L1 distance). We are also aware of another ICLR 2021 paper dealing with CATER [4], which will also be included in our revision.
>
> 3. "Is there a pretraining phase for the masked object representation prediction? Or the auxiliary loss is applied together with the classification loss?"
>
> The auxiliary loss is applied together with the classification loss; there is no pretraining. This will be clarified in the paper.
>
> 4. "What does it mean by 'MONet does not assign objects to slot indices in a well defined way'?"
>
> Clarified above in point 1; we will also clarify in the paper.
>
> 5. "What's the attention design for 'hierarchical attention model'? Is the representation input to transformer for the whole image rather than single object?"
>
> This will be clarified in our paper. Briefly, in “hierarchical attention”, we have two transformers. The first transformer operates on object slots in one frame. This transformer is applied independently to all frames of the video. The outputs of this first transformer for each frame are concatenated into a single feature vector, one feature vector per frame. These vectors are fed into the second transformer. As you say, the inputs for the second transformer represents the whole image rather than single objects.
>
> References:
>
> [1] K. Yi et al. CLEVRER: CoLlision Events for Video REpresentation and Reasoning. ICLR
> 2020
>
> [2] https://mitibmwatsonailab.mit.edu/research/blog/clevrer-the-first-video-dataset-for-neuro-symbolic-reasoning/
>
> [3] Grounding Physical Object and Event Concepts Through Dynamic Visual Reasoning. Under review for ICLR 2021. https://openreview.net/forum?id=bhCDO_cEGCz
>
> [4] Hopper: Multi-hop Transformer for Spatiotemporal Reasoning. Under review for ICLR 2021. https://openreview.net/forum?id=MaZFq7bJif7

---

### Official Review · AnonReviewer1 · 2020-10-29
**Difficult to understand the paper, the technical contributions are interesting**

**Rating:** 4
**Confidence:** 4

**Review:**

The paper is bad-written, very difficult to extract, and understand what the authors want to express. There are long sentences with wrong clauses and prepositions. The descriptions are very unclear, figures are not illustrative. I suggest the author polish the paper in a better form.

Technically, it combines the MONet and self-supervised learning via masking the objects and making the self-supervised learning as an auxiliary task. I agree with the motivation and observation that the high-level tasks that requires object-level understanding are similar to the BERT. The results also demonstrate the auxiliary task will benefit the self-supervised learning and improve the performance a lot.

I am curious about the relations and methods of using the self-supervised learning as an auxiliary task. Usually the auxiliary task does not directly improve the original task with a huge margin since the model has additional objectives, and methods like BERT also used the self-supervised learning for pre-training, not as auxiliary task. The paper that referred to as using auxiliary loss actually used the self-supervised learning as pre-training. I wonder what is the performance if we pre-training first and fine-tune with supervised loss.

Overall, I believe there are huge improvement space for the writing. The results look great, but I am still curious about where the significant improvements come from, since my previous experience suggest the self-supervised auxiliary task cannot usually bring huge improvement to the supervised learning tasks.

---

> ### Author Response · Authors · 2020-11-12
> **Clarification request**
>
> Thank you for your review. Can you please provide us with particular examples of bad writing? As noted in another review, our “paper is easy to follow”, so we aren’t exactly sure what we need to address from your perspective.
>
> You note that “self-supervised auxiliary task[s] cannot usually bring huge improvement[s]”. Self-supervised learning affords only modest improvements in some domains if performance is already close to saturation. But in principle there is nothing inherent to self-supervised learning, pre-trained or not, that precludes large or small performance changes. Can you please provide evidence to back up your claim, and provide rationale why such a perspective weakens our contribution?

---

### Official Review · AnonReviewer4 · 2020-11-03
**Clear message and solid execution, but experiment is limited**

**Rating:** 6
**Confidence:** 4

**Review:**

Summary: the paper propose to tackle visual reasoning problem in videos. The proposed solution is to combine MONET (Burgess et al., 2019) with self-attention mechanism (Vaswani et al., 2017) to first encode images into object-centric encodings and aggregate the encodings using self-attention to make the final prediction. The method is shown to outperform neural-symbolic reasoning approaches such as MAC (Hudson & Manning, 2018) and NS-DR (Yi et al 2020.) on image QA and R3D (Girdhar & Ramanan, 2020) on CARTER, which is a video reasoning task / benchmark.

Positives:
- The overall goal, i.e., showing that neural network can solve reasoning problem without specialized supervisions and structures, is clear and well-motivated.
- The solution chosen makes intuitive sense --- discover objects using MONET and encode object encodings to make the final predictions using transformers.

Comments:
- The proposed method is intuitive and does prove the main hypothesis of the paper in terms of benchmark performance, but I wish the paper can provide more intuition on how exactly the attention mechanism is able to perform visual reasoning. Specifically, I'd like to see either qualitative examples of minimum toy examples comparing the baseline methods and proposed method and show, clearly, under what circumstances does the proposed method perform better than baselines and why. This would make the main message of the paper even more salient and bulletproof.

---

> ### Author Response · Authors · 2020-11-16
> **Re: Intuition on attention mechanism**
>
> Thank you for your review. To address your concern about lack of intuition on how the attention mechanism works, we added a “Qualitative Analysis” section to our paper (Appendix C for now). In this section, we provide detailed analysis of attention weights for one particular video and some of its questions. We look at:
> * Cross-modal attention between words and objects. We note how objects attend upon words that describe them, such as a cylinder attending to the words “cylinder” and “removed” in the sentence “If the cylinder is removed, [...]”.
> * How the BERT CLS token (which is used to generate the final answer) attends to the objects. We observe that this attention depends on the question. The model generally focuses on objects that are about to collide or are mentioned in the question.
> * How the model achieves alignment across time. MONet does not assign objects to slots in the same order across the video; at different frames with the same objects, MONet could potentially put these objects into slots in different permutations. The transformer is able to track object identity through these switches and maintain consistent focus on a set of objects.
> * What the transformed MONet latents (the output of the transformer for each input latent) look like. In our paper, we introduced a self-supervised loss to encourage the transformer in learning better representations. We do this by masking out some of the input latents and tasking the transformer with predicting what the masked out latents are. To test our hypothesis that this is effective (beyond improvements of scores on tasks), we mask out entire frames and regenerate them using the transformer-predicted latents. We find that the transformer is indeed able to generate realistic looking frames. It can also predict the future motions of objects, although we identified significant room for improvement that we leave to future work.
>
> We hope this additional data was what you were looking for with regards to intuition on how the attention mechanism was able to perform visual reasoning. We thank you for this helpful suggestion, which has surely improved our paper. If you have any more suggestions, please let us know; we still have a week left in the rebuttal process!

---

### Author Response · Authors · 2020-11-16
**Revision Notes**

We thank all reviewers for their constructive and detailed reviews of our paper. We have uploaded our revision addressing some of your concerns, and we look forward to working with all of you to further improve our paper’s quality and clarity.

The revision contains some clarifications as requested by AnonReviewer3 and a comprehensive qualitative analysis of our results (currently in Appendix C --- please take a look), as suggested by AnonReviewer4. We have also been made aware of new results in CLEVRER and CATER reported in various papers [1-3], some of which are under consideration for ICLR 2021. These results are now included in our paper in order to ensure our baselines reflect the state of the art as accurately as possible.


[1] Grounding Physical Object and Event Concepts Through Dynamic Visual Reasoning. Under review for ICLR 2021. https://openreview.net/forum?id=bhCDO_cEGCz

[2] A. Shamsian et al (2020). Learning Object Permanence from Video. https://arxiv.org/pdf/2003.10469.pdf

[3] Hopper: Multi-hop Transformer for Spatiotemporal Reasoning. Under review for ICLR 2021. https://openreview.net/forum?id=MaZFq7bJif7

---

### Decision · Program_Chairs · 2021-01-07
**Final Decision**

**Decision:**

Reject

**Comment:**

This paper presents an approach to tackle visual reasoning by combining MONET and transformers. All reviewers agree that there is some performance improvement shown. But there are several concerns including clarity/writing (multiple reviewers point it), experiments (baselines) and most importantly missing insights from experiments (why it works). While some of the concerns have been handled in rebuttal, the paper still falls short on primary concern of insights/why it works (which reviewers argue is critical for a paper on reasoning). AC agrees that the paper is not yet ready for publication.